# Effects of Daily Matcha and Caffeine Intake on Mild Acute Psychological Stress-Related Cognitive Function in Middle-Aged and Older Adults: A Randomized Placebo-Controlled Study

**DOI:** 10.3390/nu13051700

**Published:** 2021-05-17

**Authors:** Yoshitake Baba, Shun Inagaki, Sae Nakagawa, Makoto Kobayashi, Toshiyuki Kaneko, Takanobu Takihara

**Affiliations:** 1Central Research Institute, ITO EN, Ltd., 21 Mekami, Shizuoka, Makinohara 421-0516, Japan; shu-inagaki@itoen.co.jp (S.I.); sae-nakagawa@itoen.co.jp (S.N.); m-kobayasi@itoen.co.jp (M.K.); t-takihara@itoen.co.jp (T.T.); 2Tokyo Skytree Station Medical Clinic, Ryobi Building F4 33-13, Mukojima 3-chome, Sumida-ku, Tokyo 131-0033, Japan; t.kaneko.19790402@gmail.com

**Keywords:** matcha, green tea, caffeine, cognitive function, attention, middle-aged and older adults, randomized placebo-controlled trial, mild acute psychological stress, executive function, performance

## Abstract

Matcha, a type of green tea, has a higher amino acid content than other types of tea. We previously examined the ability of matcha to improve cognitive function in older adults and determined that continuous matcha intake improves attention and executive function. This study aimed to compare the effects of matcha and caffeine and clarify the differences between these effects. The study was registered at the University Hospital Medical Information Network (UMIN000036578). The effect of single and continuous intake was compared, and the usefulness of continuous intake was evaluated under the stress condition. The Uchida–Kraepelin test (UKT) was used to induce mild acute stress, and the Cognitrax was used to evaluate cognitive function. A single dose of caffeine improved attentional function during or after stress loading. The reduced reaction time in the Cognitrax, observed following a single dose of matcha, was likely due to caffeine. The matcha group showed an increase in the amount of work after continuous intake, whereas the caffeine group only showed an increase in the amount of work for the UKT after a single dose. Ingesting matcha with caffeine improves both attention and work performance when suffering from psychological stress compared with caffeine alone.

## 1. Introduction

Tea is a beverage that is consumed worldwide, and green tea accounts for 1/4 of tea sales [1]. Green tea contains caffeine. The effects of caffeine on performance and mood have been extensively evaluated [2]. Animal studies have shown improved chronic stress-related behavior [3]. In humans, caffeine intake has been shown to shorten reaction time and reduce mental fatigue in cognitive function tests [4]. It has also been shown to enhance physical performance [5,6].

There are several mechanisms of action by which caffeine affects psychological function. Caffeine blocks the adenosine A_2A_ receptor, resulting in the secondary promotion of the excitatory neurotransmitters released into the synaptic cleft [2]. Adenosine receptors regulate the release of the excitatory neurotransmitter glutamate. When caffeine blocks adenosine binding to the adenosine receptor, nerve cells become excited. This is thought to be caused by exciting neural circuits that project from the basal ganglia to the striatum and cerebral cortex [7]. High concentrations of caffeine reduce and inhibit phosphodiesterase activity, which is required to regulate neural activity [2]. It also stimulates dopamine D2 receptors in the brain. Stimulation of dopamine D2 receptors occurs indirectly through the blockade of adenosine receptors, especially A2A receptors [7]. It is thought to affect many psychological functions [2]. Thus far, we have shown that continuous matcha intake may slow the decline in cognitive function in older adults [8]. We assumed that this effect also included the effects of caffeine. However, it is not clear which cognitive functions caffeine influences, and studies have not elucidated whether the effects of single component caffeine and complex component matcha differ.

Green tea is made from tea leaves (*Camellia sinensis* L.). Among the different types of green teas, matcha is characterized by its ingredients. To block sunlight and increase the amino acid content, the tea leaves used for matcha are covered for approximately 20 days before they are plucked [9]. Matcha that is made using this method has a higher amino acid content than sencha and, in general, matcha has a lower catechin content than sencha [8]. After they are harvested, the dried leaves are ground into a fine powder.

A characteristic ingredient of green tea is theanine, a non-protein amino acid that only exists in tea and mushrooms [10]. Theanine has anti-stress effects. Animal studies on theanine have shown improved behavior under stressful conditions [11] and improved learning ability under socially stressful conditions [12]. Human studies demonstrated a decrease in heart rate and salivary secretory immunoglobulin A (s-IgA), a type of immunoglobulin in the mucous membrane, when participants were exposed to acute stress [13]. Theanine also has beneficial effects on cognitive function. Studies on a mice model that promoted aging reported improvements in cognitive function [14] and the promotion of neurogenesis in the developing hippocampus [15]. In humans, theanine results in improved attention and reaction rates in response to cognitive tasks [8,16], as well as decreased α-amylase activity [17]. Theanine ingestion affects dopamine, serotonin [18], tryptophan, and gamma-aminobutyric acid [19] levels in the brain. Furthermore, theanine inhibits glutamine transport in neurons and astroglia in rat brains [20], which is presumed to alleviate glutamic acid excitotoxicity. Excitatory neuronal toxicity is thought to be involved in neurodegenerative diseases, such as Alzheimer’s disease (AD) [21]. A previous report [8] found that continuous intake of matcha containing a high theanine concentration improved cognitive function.

Animal studies have reported that catechins suppress amyloid beta peptide (Aβ) production. The production of Aβ is thought to be one of the factors associated with cognitive impairment in AD [22,23,24]. Therefore, we investigated the effects of consuming 336.4 mg of catechins for 12 weeks on Aβ and its related proteins, Aβ precursor protein α (sAPPα) and Aβ precursor protein 770 (APP770). Furthermore, serum brain-derived neurotrophic factor (BDNF) levels were measured to examine cognitive function [25]. The animal studies had problems that needed to be addressed, such as the administration period and blood evaluations. Thus, in this study, these markers were continuously measured as secondary endpoints to collect the background data of healthy middle-aged and older adults and examine matcha’s effects.

Interestingly, green tea contains theanine and caffeine, which simultaneously affect neurotransmitters. Caffeine [26] acts as a stimulant for neural activity, while L-theanine [27] acts as a depressant. Beneficial effects have also been demonstrated through their interaction [28]. Catechin, a polyphenol of green tea [29,30], also affects cognitive function. The overall effects of the three components (caffeine, theanine, and catechin) are thought to be the reason for matcha’s ability to improve cognitive function. However, it is still unclear which differences in cognitive function are caused by caffeine alone and which are caused by the complex components of caffeine, theanine, and catechin.

Cognitive function declines with age [31]; however, aside from age-related cognitive function decline, fatigue [32,33] and stress [34] also cause cognitive decline. Mental tasks, even performed over a short period, cause stress. Performing the 2-back test, which is used to evaluate working memory for 30 min, stimulates the autonomic nervous system by allowing it to reach the same level of fatigue as in daily life [35,36]. Fatigue deteriorates the quality of life [37]. Studies have shown that if one cannot recover from fatigue, it can have harmful effects on one’s social life [38,39]. Thus far, we have shown that continuous matcha and catechin intake [8,25] may suppress cognitive function decline in older adults. However, the effects of stress on cognitive function in older adults are unclear. The Uchida–Kraepelin test (UKT) is used to assess mild acute psychological stress. Although a significant increase in 3-methoxy-4-hydroxyphenylglycol, a metabolite of noradrenaline [40], was not noted when participants performed the 30 min task, participants reported increased subjective fatigue [41]. The UKT is widely used by Japanese companies during the recruitment process as an aptitude test. Examinees’ character and work capacity are evaluated by the change in the number of operations they can accomplish while adding a series of single-digit numbers [40]. In previous studies, we established that matcha may maintain attention during or after stress in younger adults [42]. To improve quality of life and delay cognitive decline, it is important that the effects of matcha on mild stress in older adults be elucidated and that effective foods be determined.

The purpose of this study was to clarify whether the same amount of caffeine in matcha contributed to cognitive function improvements and to investigate which cognitive functions were affected. Furthermore, we also investigated whether matcha had a beneficial effect on cognitive function in middle-aged and older adults exposed to mild, acute psychological stress.

## 2. Materials and Methods

### 2.1. Ethics Statement

The study protocol was approved by the ethics committee of Nihonbashi Egawa Clinic (Tokyo, Japan; approval number: food-19031102). The study was conducted in accordance with the tenets of the Helsinki Declaration (adopted in 1964, amended in 2013). The study was conducted at the Tokyo Skytree Station Medical Clinic (Tokyo, Japan) from April 2019 to August 2019. The study was registered at the University Hospital Medical Information Network (UMIN; Tokyo, Japan; number, UMIN000036578).

### 2.2. Test Food

Hojin no shiro (ITO EN, Ltd., Tokyo, Japan) was used as the matcha. The total catechin, theanine, and caffeine contents in the daily matcha intake (2070 mg) are indicated in Table 1. Catechins (total) is the total value of epigallocatechin gallate (EGCG), gallocatechin gallate (GCG), epicatechin gallate (ECG), catechin gallate (CG), epigallocatechin (EGC), gallocatechin (GC), epicatechin (EC), and catechin (C). For the caffeine component, a caffeine extract (Shiratori Pharmaceutical Co., Ltd.; Chiba, Japan) was used. The daily test food intake was designed to be met through the consumption of nine capsules.

White, porcine, gelatin capsules (number 1) were used in this study. The placebo and caffeine used the same capsule as the matcha, which was starch-colored green. The matcha, caffeine, and placebo capsules used starch as the excipient. The matcha capsules were manufactured by Sunsho Pharmaceutical Co., Ltd. (Shizuoka, Japan), and the caffeine capsules were manufactured by Shefco Co., Ltd. (Tokyo, Japan).

### 2.3. Participants

We recruited healthy Japanese men and women aged 50–69 years who reported a self-assessed decline in cognitive function. The selection criteria were: (1) ability to take 9 capsules daily for 12 consecutive weeks, (2) MMSE-J score of 24 or more, and (3) non-smokers. Participants who met the following criteria were excluded from the study: (1) currently taking any medications or receiving outpatient treatments; (2) a history of or current serious disease including liver, kidney, endocrine, cardiovascular, gastrointestinal, lung, blood, or metabolic diseases and any history of, or current, complications; (3) a history of drug and/or food allergies; (4) use of health foods and/or supplements that may have affected cognitive function; (5) use of medications that may have affected cognitive function; (6) an extremely unbalanced diet; (7) extremely irregular lifestyle habits, including sleep; (8) suspected insomnia; (9) a history of psychiatric disorders, such as depression; (10) a history of alcoholism; (11) current participation in a clinical trial or participation in one within the last 3 months; (12) irregular working hours, such as night shifts; and (13) deemed as an inappropriate participant by the doctor.

The 51 participants who were selected through the screening test were randomly assigned, using a random number table, to the following 3 groups: placebo, caffeine, or matcha. The allocation factors were age, sex, and Cognitrax score. The sample size was determined based on a previous report [43,44,45,46,47]. The Cognitrax score was used to equalize the participant’s ability to perform the task. From the Cognitrax tests, the Language Memory, Visual Memory, Stroop, and 4-Part Continuous Processing tests were adopted in order from the test with the highest number of incorrect answers and slowest reaction times. The participants performed 4 Cognitrax tasks (Verbal Memory test [VBM], Visual Memory test [VIM], Stroop test [ST], and 4-part Continuous Performance test [FPCPT]), and allocations were based on the largest number of incorrect answers of the 4 total test scores and the longest total reaction times of the 4 tests. The randomization and blinded process was performed at HUMA R&D CORP. (a contract research organization) in Tokyo, Japan.

### 2.4. Study Design

We performed a double-blind, randomized, placebo-controlled, parallel-group study. The study flow diagram is shown in Figure 1. The primary endpoints were the Cognitrax and UKT results. The secondary endpoints included the serum Aβ (1–40), Aβ (1–42), sAPPα, APP770, and BDNF levels.

Participants took 9 placebo, caffeine, or matcha capsules daily for 12 weeks in the morning. Intake was recorded using a web input system (HUMA R&D CORP.). The participants did not need to restrict their polyphenol (green tea, black tea, oolong tea, etc.) intake, and took the capsule in addition to their usual diet. However, they were instructed to avoid excessive exercise, excessive dietary restrictions, binge eating, and heavy drinking. While ingesting the test foods, participants were prohibited from consuming healthy foods, supplements, or medications that may have affected their cognitive function. The participants were instructed to avoid intake of these products as much as possible; however, there were no restrictions on other types of healthy foods and supplements. The type and amount of the product were recorded when any of these products were taken. With respect to medication, the participants were required to record the name and dose of the drug. In addition to these limitations, the participants were told to maintain their normal lifestyles. Daily diary entries were completed to record their intake statuses and the presence of adverse events, such as colds, stomachaches, etc.

### 2.5. Patient Evaluations

The evaluation scheme for the clinical trial is presented in Table 2. The evaluations, which were marked with circles, were administered on the day of the test. In the single-dose study, the Cognitrax test was started approximately 50 min after capsule intake. The following restrictions were enforced before participants visited the hospital for the test: (1) participants refrained from performing a prolonged or intense exercise that caused shortness of breath from the day before the test; (2) participants avoided overeating, excessive dieting, lack of sleep, excessive exertion that deviated from normal daily activities, and alcohol consumption the day before the test; and (3) participants ensured that meals consumed on the day of the test were finished at least 6 h before the test because baseline and 12 week measurements and blood samples were taken. Only water could be consumed during the period between the end of the last meal and the end of the test.

#### 2.5.1. MMSE-J

The MMSE-J (Nihon Bunka Kagakusha Co., Ltd., Tokyo, Japan), the Japanese version of the MMSE, was used in this study. Its validity and reliability have been confirmed in Japan [48]. The test consists of 11 items: time orientation, location orientation, memorization, attention and calculation, recall, naming, repetition, 3-stage command, reading, writing, and copying. It is evaluated using the total score. The score for the backward spelling task was used for participant allocation.

#### 2.5.2. The Cognitrax Test

Cognitrax [49] was used as a test battery for cognitive function. The Cognitrax tests were performed in the following order: VBM, VIM, Finger Tapping test (FTT), Symbol Digit Coding test (SDC), ST, Shifting Attention test (SAT), Continuous Performance test (CPT), Perception of Emotions test (POET), Non-Verbal Reasoning test (NVRT), FPCPT, VBM, and VIM. The results of the first VBM and VIM tests were described as immediate memory, and the results of the last VBM and VIM tests were described as delayed memory. There was an interval of approximately 50 min between the first and last VBMs and VIMs. Performances on the memory, attention, facial expression recognition, working memory, visual information processing, and motor function tasks were determined using the VBM and VIM; ST, SAT, CPT, and FPCPT parts 1 and 2; POET; FPCPT parts 3 and 4; SDC and NVRT; and FTT, respectively. The outline of each task was as follows:VBM: First, participants memorized 15 words. Then, from the 30 words that appeared at random, they selected the words that they remembered.VIM: A graphic version of the VBM.FTT: Participants quickly tapped a key for 10 s with their index finger. Both the right and left hands were evaluated.SDC: Participants entered the number that corresponded to the symbol while referring to the symbol-to-number correspondence table.ST: Consisted of parts 1 to 3. Part 1 involved pressing the key when a character appeared. Part 2 involved pressing a key when the meaning of the letters and colors matched. Part 3 involved pressing a key when the letters and colors did not match.SAT: Participants followed the on-screen instructions and selected the option that matched the color or shape. There was a combination of red, blue, yellow, and green letters and colors.CPT: The alphabet was randomly displayed one by one. Participants were instructed to only press the key when B was displayed.POET: A face photo and facial expressions were displayed. The facial expression was displayed with 4 words, “calm”, “happy”, “sad”, and “angry”. Participants were instructed to only press the key if the photo and description matched. We evaluated positive (calm and happiness) and negative emotions (sadness and anger). The results showed the following 3 values: the sums of the positive and negative emotions, positive emotions alone, and negative emotions alone.NVRT: One of the 4 sections was blank and 3 had symbols. Participants were instructed to select the blank symbol from the 5 examples.FPCPT: Consisted of parts 1 to 4. Part 1 involved pressing the key when the figure appeared. Part 2 involved pressing the key when a green circle appeared. Part 3 was a 1-back task. Part 4 was a 2-back task. Combinations of circle, triangle, square, and star figures and red, blue, yellow, and green colors were provided.

#### 2.5.3. UKT

The UKT involved a series of additions of single-digit numbers [40]. The answer sheet consisted of 34 lines. Each line presented 115 addition operations that required completion. Participants were instructed to move to the next line each minute from the start of the test. The test consisted of 2 blocks, each lasting 15 min. After completing the first session, the participants were allowed to rest for 5 min. The total duration of the test was 30 min. The first 15 min was regarded as the first block, and the next 15 min was regarded as the second block.

The average number of operations completed was the average number of answers per line over 30 min. The 3-line number of errors was calculated by taking one line from the first half of the test and 2 lines from the second half of the test and summing the number of errors in all 3 lines.

### 2.6. Statistical Analysis

The values were reported as the mean ± standard deviation (SD). Normality was tested using the Shapiro–Wilk test. Levene’s test was used to test for homoscedasticity. For the groups, the Kruskal–Wallis test followed by a post-hoc Mann–Whitney *U* test was used at each time point to test for inter-group outcomes. The *p*-value was determined using a Bonferroni correction of *p* < 0.05/3 = 0.017 (control vs. caffeine, control vs. matcha, and caffeine vs. matcha). Regarding the times, Dunnett’s test was used for intra-group comparisons, with the baseline condition as the reference. The *p*-value was determined using a Bonferroni correction of *p* < 0.05/2 = 0.017 (baseline vs. 0 W, baseline vs. 12 W). Data were analyzed using SAS version 9.4 (SAS Institute Inc.; Cary, NC, USA).

## 3. Results

The participants’ clinical characteristics are shown in Table 3. Participants were free to consume polyphenol-containing tea and beverages during the study period. Before the intervention, we investigated the amount of green tea and coffee that the participants consumed in their daily lives. Regarding green tea, the placebo, caffeine, and matcha groups consumed 7.07 ± 8.8 cups/week, 4.90 ± 5.0 cups/week, and 9.27 ± 8.5 cups/week, respectively. Regarding coffee, the placebo, caffeine, and matcha groups consumed 13.1 ± 9.1 cups/week, 8.71 ± 8.6 cups/week, and 8.12 ± 5.6 cups/week, respectively. No significant differences were observed between the 3 groups.

### 3.1. MMSE-J

The average scores from the Japanese version of the mini-mental state examination (MMSE-J; backwards task) before the intervention (baseline) were 28.7 ± 0.8, 28.2 ± 1.2, and 28.4 ± 1.8 in the placebo, caffeine, and matcha groups, respectively. Post-intervention, the average scores were 26.5 ± 2.8, 28.5 ± 1.7, and 28.6 ± 1.4 in the placebo, caffeine, and matcha groups, respectively. After the intervention, the caffeine (*p* = 0.016) and matcha groups showed significantly higher results than the placebo group (baseline: Kruskal–Wallis test [KW], *p* = 0.04; 12 weeks: KW, *p* = 0.01). The placebo group had a significantly lower result at 12 weeks compared to that at the baseline (Mann–Whitney *U* test, *p* = 0.009).

### 3.2. Cognitrax Test

#### 3.2.1. Memory Tasks

Matcha and caffeine had no effects on the VBM and VIM (Table 4).

#### 3.2.2. Attention Tasks

With respect to the SAT, the amount of change from the baseline at a single dose, the caffeine group had a significantly faster reaction time providing the correct answer compared to the placebo group. Regarding the POET and the amount of change from 0 weeks at a single dose, the caffeine and matcha groups had significantly faster-corrected reaction choice times than the placebo group. Moreover, with respect to FPCPT (part 2) and the amount of change from 0 weeks at a single dose, the caffeine group had a significantly faster corrected average response time than the placebo group. In the matcha group, there were significantly more incorrect responses after a single dose compared with the placebo group, and the average incorrect response time also increased significantly compared to the placebo group. The amount of change from the baseline after a single dose was significantly lower in the matcha group than the caffeine group (Table 5).

#### 3.2.3. Facial Expression Recognition Task

Caffeine and matcha consumption did not affect the POET (Table 6).

#### 3.2.4. Working Memory Tasks

In part 3 of the FPCPT, matcha, and caffeine did not affect the 1-back tasks. However, in part 4, the caffeine group had a significantly faster average correct response time than the placebo group after a single dose. With respect to the incorrect response times and the amount of change from 0 weeks at a single dose, the caffeine group had a significantly faster average correct response time than the placebo group (Table 7).

#### 3.2.5. Visual Information Processing Tasks

Matcha and caffeine had no effects on the SDC and NVRT (Table 8).

#### 3.2.6. Motor Function Tasks

Matcha and caffeine had no effects on the FTT (Table 9).

### 3.3. The UKT

Regarding the first block score and the percentage of change from baseline at 0 W, the caffeine group’s scores were significantly higher than the placebo group. Contrastingly, regarding the total, first block, and average scores, and percentage change from the baseline at 12 weeks, the matcha group had a significantly higher score than the placebo group (Figure 2). The KW *p*-value for the number of errors in the 3 lines after the single dose was 0.03.

### 3.4. Blood Biomarkers

Plasma Aβ (1–42) at 12 weeks showed a significant difference in the KW test. However, no significant difference was found in the post-hoc test between the placebo, caffeine, and matcha groups. There were no significant differences in the plasma for the levels of Aβ1–40, Aβ1–40/Aβ1–42 ratio, sAPPα, and APP770, or in the serum for BDNF levels, in the KW test (Table 10).

## 4. Discussion

This study aimed to clarify the role that caffeine plays in the capacity of matcha to improve cognitive function and to assess whether matcha has a beneficial effect on the cognitive decline caused by mild acute psychological stress among middle-aged and older adult groups. Previous studies on catechins [25] and theanine [50] found that their effects differed depending on the intake period. Therefore, the effects of a single administration and continuous intake for 12 weeks were also examined in this study. The result suggests that caffeine in matcha contributed to the extension of reaction time of cognitive function tasks. However, matcha showed an increase in the amount of work for the simple calculation task, and a beneficial effect for the performance under stress load, which was better than that of caffeine.

Previous studies have revealed that continuous matcha intake improves attention and executive function in older adults [8], suggesting that matcha may suppress the cognitive function decline associated with aging. Caffeine is a component of matcha whose effects on performance and mood have been extensively evaluated [2]. The combination of caffeine and theanine contributes to improved attention [28]. Matcha, which simultaneously contains caffeine and theanine, is presumed to have a better effect on cognitive function than caffeine alone.

In addition to the age-related decline in cognitive function, fatigue also causes cognitive decline [32]. Moreover, depending on its quality, quantity, and duration, stress can affect cognitive function [51]. Studies have suggested that matcha may maintain attention during and after attention-demanding cognitive function tasks in young participants [42]. However, the effect of age-related cognitive function decline is unknown. Therefore, in this study, we examined matcha’s effects under stressed conditions in middle-aged and older adult participants.

As a result of a single dose of caffeine equivalent to that contained in matcha, we found that the SAT, CPT, and FPCPT (parts 2 and 4) reaction times were shortened. Ingesting caffeine led to a quicker response for both when providing the correct answer and when providing the wrong answer (Table 7 FPCPT [part 4]). Haskell et al. reported that when 150 mg of caffeine was administered to participants with an average of 21.3 ± 0.83 standard error of the mean, the digit vigilance reaction time was faster [52] and the reduction in reaction time tended to be the same as that found in this study and as previously reported. In the caffeine group, the reaction time was shortened in many parameters, whereas in the matcha group, this was only observed in the CPT. This may be due to differences in the amounts of caffeine and theanine that were administered. Haskell et al. [28] reported that the theanine level was approximately 1.6 times higher than that for caffeine. In our study, the theanine level was approximately 0.7 times the caffeine level. Furthermore, in the study by Haskell et al., the amount of theanine was approximately five times that in our test, and their study used approximately twice as much caffeine. According to Kakuda et al. [27], theanine diminishes the excitatory effects of caffeine. However, future studies on the ratio of theanine to caffeine and the amounts of theanine and caffeine are still required.

Contrastingly, Kobayashi et al. [53] reported that the anxiety group did not show alpha waves at 50 mg of theanine; however, the high anxiety group did. Our study did not investigate mental status; however, it was assumed that the participants were experiencing stress. To simulate mild acute psychological stress, the UKT was performed for 30 min. In studies that used the UKT, increased noradrenaline nerve activity [40], fatigue, and increased chromogranin [41] were observed, and it is speculated that sympathetic nerve activity was activated. Therefore, we believe that theanine in matcha might improve cognitive function due to its anti-stress effect; however, these effects were not observed in the cognitive function tests in the current study. The matcha group had fewer tests with shorter reaction times than the caffeine group. We speculate that the cause of this was the difference in cognitive function between young and middle-aged participants. Both Haskell et al. and Kobayashi et al.’s studies included participants in their twenties. When investigating younger adults, the administration of matcha may help to maintain attention [42]. There is much scope for future research on the topic of age-related cognitive decline, especially on how to maintain attention. We propose that the participants’ ages must be considered in future studies.

In this study, participants with an MMSE score of 24 points or higher were considered as those without suspicion of dementia. The MMSE score for the placebo group decreased from 28.7 to 26.5. The MMSE score was 27/28 for healthy participants with mild cognitive impairment (MCI) and 23/24 for MCI with mild AD [54]. The placebo group declined from the healthy range to the MCI range. In the placebo group, the scores of three participants were 28–22, 28–23, and 28–21. In other words, the patients transitioned from within a healthy range to mild dementia within 12 weeks. Recalls were the lowest in the MMSE, which consisted of three words: a ball, a flag, and a cherry blossom. When assessed using Cognitrax (CNS Vital Signs, LLC; Morrisville, NC, USA), no deterioration in performance was observed in the placebo group (Table 4, VBM correct hits [delayed]). Thus, it was unlikely that the patients developed mild dementia. Since this study did not include diagnoses made by dementia specialists, it is unclear whether the patients’ cognitive function truly shifted into the range of mild dementia.

Regarding the effects of theanine, Kakuda et al. [27] indicated that theanine alone has different effects depending on its concentration. In an in vivo study using electroencephalography, an equimolar concentration or more of theanine was required to eliminate caffeine’s effect. This principle has not yet been demonstrated in humans, and theanine levels were lower than caffeine levels in this study. Therefore, it was inferred that the effect of caffeine was not counteracted. Kakuda et al. also noted that small amounts of theanine had an excitatory effect. Theanine has interesting effects when influenced by caffeine and should be investigated in future studies. It is likely that the ratio of caffeine to theanine in the presence of caffeine also influenced this effect in this study.

Theanine is a green tea ingredient that exhibits anti-stress activity. Animal studies have reported that cortisol is reduced after stress loading [13] and that the increase in heart rate and s-IgA caused by restraint stress is suppressed [55]. Theanine acts as a partial agonist to the *N*-methyl-*D*-aspartate receptor, a glutamine receptor [56]. Kobayashi et al. measured human alpha waves and found that the alpha wave, an index of relaxation, did not appear at 50 mg, and that 200 mg was required to see an effect [53]. This study used 50 mg of theanine; thus, it is possible that there was no relaxing effect. The difference between this study and that by Kobayashi et al. is the presence of caffeine. Theanine’s effects are considered to be derived from the ratio of caffeine to caffeine when it is present, and its concentration when it is not present. Further studies are needed regarding the caffeine ratio and the amount that should be administered. In the caffeine group, the UKT results showed a significant increase in the first block score after a single dose was administered (daily intake, *p* < 0.0065). The matcha group showed high values following a single dose (*p* < 0.0175). On the other hand, with long-term administration, the total score and first block score of theanine alone were significantly higher than those of the placebo group. This indicates that continuous intake of matcha contributes to the attention maintenance during the continuous single-digit addition trials through phenomena such as theanine’s anti-stress effect.

Daily intake of catechins, which are contained in green tea, may also affect cognitive function [25]. Catechins hold the potential to remove free radicals and protect cells [57,58]. Since the brain is vulnerable to oxidative stress, the catechins’ antioxidant action is considered to play an important role. According to a report by Unno et al. caffeine-free green tea catechin extract improves learning ability and cognitive function [59]. EGCG metabolites also have antioxidants and positive effects on nerve cells [60]. The matcha group had high UKT scores after 12 weeks of continuous intake and we inferred that this was due to the catechins. In addition to examining the interaction between caffeine and theanine, caffeine, theanine, and catechins should also be examined.

This study examined plasma Aβ (1–40), Aβ (1–42), APP770, sAPPα, and serum BDNF levels in relation to cognitive function disorders in healthy middle-aged and older participants. In previous studies, we measured the changes in the aforementioned markers when catechins [25] or theanine [50] were ingested for 12 weeks. In this study, we also examined whether the values changed when caffeine or matcha was ingested. In our study, we did not observe any effects from continuous matcha or caffeine intake or any changes from baseline; combined with the results so far [25,43], 12 weeks of catechin, theanine, caffeine, and matcha intake did not change these markers. Further research is required to assess cognitive function using these serum markers. However, we propose that it is important that background Aβ marker levels in healthy middle-aged and older participants without cognitive impairment be measured in future studies.

This study had some limitations, including the participants’ ages and nationality, as well as the quality of stress. The effects observed in this study were limited to Japanese participants aged 50–69 years, who had a habit of drinking green tea. Moreover, the anti-stress effect that was revealed in this study was for single-digit continuous calculations. Neither biomarker measurements, nor subjective assessments of the quality of stress or how stressed an individual was, were performed.

In conclusion, different effects were noted between the caffeine and matcha groups. A reduced cognitive function test reaction time was observed with a single dose of caffeine. Contrastingly, in the UKT, the matcha group demonstrated an increase in the amount of work performed after a single dose. In addition, the reaction time of the Stroop test improved from 12 weeks of consuming matcha, demonstrating the importance of the continuous intake of matcha. Anti-stress effects of matcha might contribute to an increase in the number of simple tasks that required attention, i.e., the maintenance or improvement of attention. Caffeine accelerated the reactions to work. This study suggested that matcha that contained caffeine was less effective at shortening the reaction time; however, we think that this may be due to a difference between the effects of matcha and caffeine, and that matcha intake increased simple workload and improved executive function. Our study suggests that theanine, catechin, and caffeine alone exert different effects than when they are present simultaneously. In the future, studies should examine how the intake of these components affects brain activity and why their ingestion leads to improvements in attention and executive function. The active site that uses fMR should be verified to confirm that the function is expressed. The authors would like to express interest in examining the activity that the simultaneous presence of theanine, catechin, and caffeine elicits in brain functioning.

## Figures and Tables

**Figure 1 nutrients-13-01700-f001:**
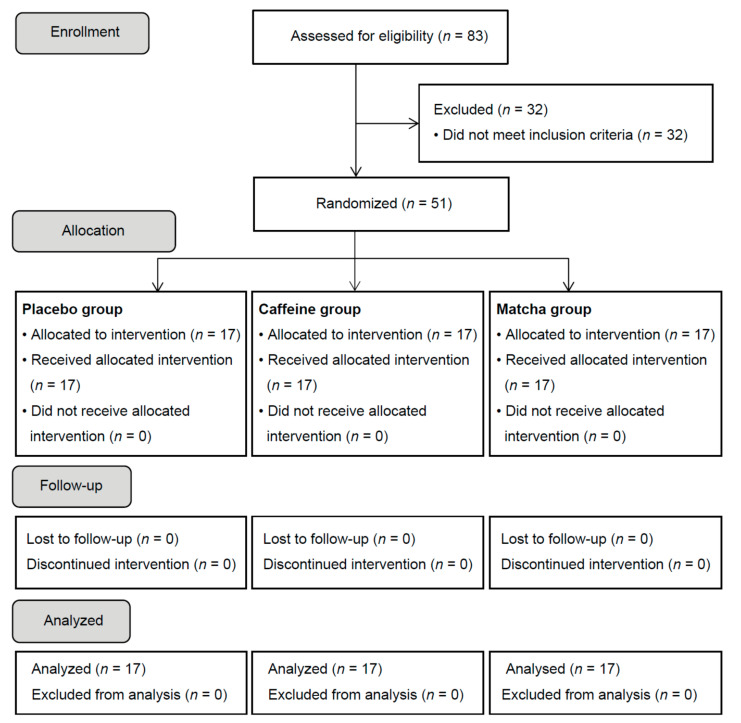
Study flow diagram.

**Figure 2 nutrients-13-01700-f002:**
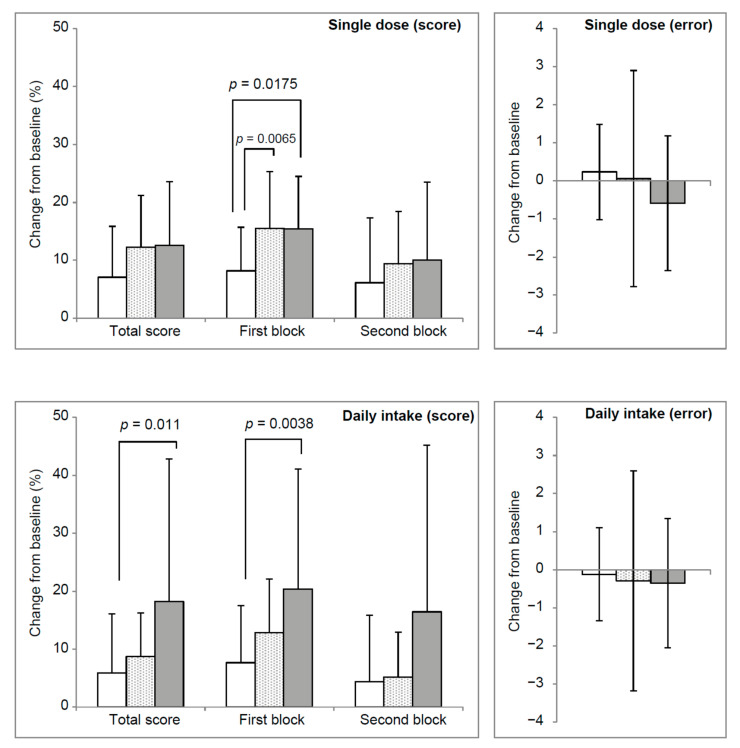
Number of answers and errors during a mild acute stress task and intake of matcha or caffeine. Shown are the placebo group (white bar), caffeine group (shadow bar), and matcha group (grey bar). Each bar shows the mean ± standard deviation. Group (inter) analysis was performed using the Kruskal–Wallis test (KW) followed by a post-hoc Mann–Whitney U test at each time point to test for inter-group outcomes (control vs. caffeine, control vs. matcha, and caffeine vs. matcha). The *p*-value was determined using a Bonferroni correction of *p* < 0.05/3 = 0.017; *p* < 0.01/3 = 0.0033 vs. placebo group.

**Table 1 nutrients-13-01700-t001:** Composition of main ingredients in test foods.

	Placebo Group	Caffeine Group	Matcha Group
Theanine (mg)	0.0	0.0	48.0
Caffeine (mg)	0.0	66.6	66.2
Total Catechins (mg)	0.0	0.0	170.0
EGCG (mg)	0.0	0.0	105.6
GCG (mg)	0.0	0.0	1.4
ECG (mg)	0.0	0.0	20.3
CG (mg)	0.0	0.0	0.1
EGC (mg)	0.0	0.0	33.1
GC (mg)	0.0	0.0	1.6
EC (mg)	0.0	0.0	8.1
C (mg)	0.0	0.0	0.8

The data show the content of the main component of the test food in the daily intake. The daily test food intake was designed to be met through the consumption of nine capsules. Abbreviations: EGCG, epigallocatechin gallate; GCG, gallocatechin gallate; ECG, epicatechin gallate; CG, catechin gallate; EGC, epigallocatechin; GC, gallocatechin; EC, epicatechin; C, catechin.

**Table 2 nutrients-13-01700-t002:** Clinical trial evaluation scheme.

	−4 Weeks	0 Weeks	12 Weeks
	(baseline)	(single dose)	
Ingestion on the test day		⬤	
Medical interview	⬤	⬤	⬤
Height	⬤		
Weight	⬤	⬤	⬤
Vital Signs	⬤	⬤	⬤
Blood sampling *	⬤		⬤
Uchida–Kraepelin Test	⬤	⬤	⬤
Cognitrax test	⬤	⬤	⬤

The evaluations that are marked with circles (⬤) were performed on the day of the test. Tests were performed in the order listed in the table. * Blood samples were collected for hematological tests (white blood cell and red blood cell counts, hemoglobin concentration, hematocrit, platelet count, mean corpuscular volume, mean corpuscular hemoglobin, and mean corpuscular hemoglobin concentration) and biochemical blood parameter evaluations (total protein, triglycerides, total cholesterol, high-density lipoprotein cholesterol, low-density lipoprotein cholesterol, alkaline phosphatase, aspartate aminotransferase, alanine aminotransferase, gamma-glutamyl transpeptidase, lactate dehydrogenase, uric acid, urea nitrogen, total bilirubin, albumin, creatinine, fasting blood glucose, and glycated hemoglobin).

**Table 3 nutrients-13-01700-t003:** The participants’ clinical characteristics.

	Placebo	Caffeine	Matcha	*p*-Value
Number of participants	17	17	17	
Sex (Male/Female)	7/10	8/9	8/9	1.00 (F)
Age (years)	58.3 ± 5.9	57.9 ± 6.4	58.3 ± 4.9	1.00 (KW)
Height (cm)	160.9 ± 7.5	162.6 ± 8.6	163.4 ± 9.5	0.77 (KW)
Weight (kg)	54.1 ± 9.5	59.3 ± 10	57.6 ± 13	0.41 (KW)
BMI	20.8 ± 2.8	22.3 ± 2.3	21.2 ± 3.0	0.28 (KW)

Values are presented as mean ± SD; *p*-values were calculated using Fisher’s exact test or the Kruskal–Wallis test. Abbreviations: SD, standard deviation; BMI, body mass index; F, Fisher’s exact test; KW, Kruskal–Wallis test.

**Table 4 nutrients-13-01700-t004:** Effect of matcha or caffeine intake on memory-related tasks.

Task			−4 Weeks (1)	0 Weeks (2)	12 Weeks (3)	Change from Baseline		*p*-Values (KW Test)				
			(Baseline)	(Single Dose)		0 W (4)	12 W (5)	(1)	(2)	(3)	(4)	(5)
VBM	Correct hits	Placebo	10.6 ± 2.2	12.5 ± 2.4	12.9 ± 2.5 #	1.94 ± 2.4	2.35 ± 1.9	0.87	0.71	0.95	0.95	0.87
	(immediate)	Caffeine	10.8 ± 2.2	12.9 ± 2.1 #	13.1 ± 1.7 #	2.12 ± 2.5	2.24 ± 2.7					
		Matcha	10.8 ± 2.1	12.4 ± 2.0	13.1 ± 1.7 ##	1.59 ± 2.6	2.29 ± 2.1					
	Correct passes	Placebo	14.6 ± 1.2	14.6 ± 0.8	14.5 ± 0.6	0.06 ± 1.5	−0.06 ± 1.3	0.42	0.87	0.56	0.78	0.40
	(immediate)	Caffeine	14.5 ± 0.7	14.5 ± 0.9	14.4 ± 0.9	0.00 ± 1.0	−0.18 ± 1.1					
		Matcha	14.6 ± 0.7	14.7 ± 0.6	14.6 ± 0.9	0.12 ± 0.7	0.00 ± 0.9					
	Correct hits	Placebo	9.06 ± 3.1	10.5 ± 3.3	11.1 ± 3.0	1.47 ± 3.7	2.00 ± 3.0	0.75	0.72	0.78	0.14	0.25
	(delayed)	Caffeine	9.12 ± 2.3	11.3 ± 2.2 #	11.0 ± 2.9	2.18 ± 2.8	1.88 ± 3.9					
		Matcha	8.59 ± 2.9	11.6 ± 2.3 #	11.7 ± 2.6 #	3.00 ± 2.4	3.12 ± 2.6					
	Correct passes	Placebo	14.1 ± 1.3	14.2 ± 1.0	14.5 ± 1.1	0.18 ± 1.1	0.41 ± 1.9	0.55	1.00	0.54	0.64	0.36
	(delayed)	Caffeine	14.4 ± 1.0	14.3 ± 0.8	14.2 ± 1.0	−0.12 ± 1.4	−0.18 ± 1.6					
		Matcha	14.2 ± 1.0	14.2 ± 1.0	13.8 ± 1.9	0.06 ± 0.8	−0.35 ± 1.5					
VIM	Correct hits	Placebo	10.3 ± 1.9	11.3 ± 2.0	11.5 ± 2.0	1.00 ± 2.0	1.18 ± 1.7	0.97	0.96	0.39	0.94	0.18
	(immediate)	Caffeine	10.5 ± 2.2	11.5 ± 2.2	11.4 ± 2.0	1.00 ± 3.2	0.88 ± 2.2					
		Matcha	10.4 ± 2.2	11.2 ± 3.4	10.5 ± 2.4	0.82 ± 3.0	0.12 ± 1.4					
	Correct passes	Placebo	12.1 ± 2.4	11.6 ± 2.2	11.5 ± 2.4	−0.47 ± 1.5	−0.59 ± 2.6	0.83	0.16	0.65	0.15	0.68
	(immediate)	Caffeine	12.6 ± 2.0	10.8 ± 2.7	11.9 ± 2.5	−1.82 ± 3.7	−0.71 ± 1.9					
		Matcha	12.5 ± 2.2	12.4 ± 2.3	12.1 ± 2.4	−0.06 ± 1.4	−0.35 ± 2.2					
	Correct hits	Placebo	9.59 ± 2.3	10.7 ± 2.6	10.4 ± 2.4	1.12 ± 2.7	0.82 ± 2.0	0.85	0.56	0.92	0.27	0.62
	(delayed)	Caffeine	9.47 ± 3.0	10.2 ± 3.1	10.4 ± 2.8	0.71 ± 4.3	0.88 ± 3.8					
		Matcha	9.94 ± 2.2	9.71 ± 2.4	10.1 ± 2.7	−0.24 ± 2.4	0.12 ± 3.2					
	Correct passes	Placebo	11.3 ± 3.0	10.9 ± 3.3	9.77 ± 2.7	−0.35 ± 2.8	−1.53 ± 2.5	0.92	0.74	0.42	0.47	0.86
	(delayed)	Caffeine	11.5 ± 2.6	10.4 ± 3.1	10.4 ± 2.7	−1.12 ± 4.0	−1.12 ± 2.6					
		Matcha	11.1 ± 2.9	10.7 ± 2.8	9.47 ± 2.4	−0.41 ± 3.2	−1.65 ± 2.4					

Values are presented as the mean ± standard deviation. Group (inter) analysis was performed using the Kruskal–Wallis test (KW). (1) (2) (3) (4) (5) The numbers in brackets: at −4 weeks, 0 weeks and 12 weeks corresponded to the numbers listed in the column with the *p*-value for the KW test. Time (intra) was analyzed using Dunnett’s test (T3) for intra-group comparisons, using the baseline condition as the reference (baseline vs. 0 W and baseline vs. 12 W). The *p*-value was determined using a Bonferroni correction of # *p* < 0.05/2 = 0.025 and ## *p* < 0.01/2 = 0.005. Abbreviations: KW, Kruskal–Wallis test; VBM, verbal memory test; VIM, visual memory test; W, weeks.

**Table 5 nutrients-13-01700-t005:** Effect of matcha or caffeine intake on attention-related tasks.

Task			−4 Weeks (1)	0 Weeks (2)	12 weeks (3)	Change from Baseline		*p*-Values (KW Test)				
			(Baseline)	(Single Dose)		0 W (4)	12 W (5)	(1)	(2)	(3)	(4)	(5)
ST (part 1)	Simple reaction time	Placebo	341 ± 40	336 ± 48	333 ± 51	−5.29 ± 40	−8.00 ± 46	0.50	0.78	0.42	0.14	0.67
	(ms) [A]	Caffeine	343 ± 72	333 ± 55	341 ± 42	−10.0 ± 59	−2.18 ± 67					
		Matcha	410 ± 228	327 ± 35	356 ± 54	−82.8 ± 203	−52.5 ± 206					
(part 2)	Complex reaction time,	Placebo	719 ± 81	687 ± 72	674 ± 82	−32.4 ± 57	−44.8 ± 66	0.15	0.29	0.58	0.52	0.11
	correct (ms) [B]	Caffeine	661 ± 91	648 ± 78	676 ± 103	−13.0 ± 76	15.6 ± 113					
		Matcha	716 ± 180	655 ± 58	638 ± 44	−60.9 ± 161	−77.6 ± 174					
(part 3)	Stroop reaction time,	Placebo	811 ± 88	766 ± 94	755 ± 78	−45.2 ± 92	−55.9 ± 8.0	0.38	0.65	0.93	0.98	0.21
	correct (ms) [C]	Caffeine	770 ± 96	732 ± 80	759 ± 104	−38.2 ± 54	−11.6 ± 77					
		Matcha	825 ± 169	761 ± 101	751 ± 85	−63.6 ± 122	−74.3 ± 130					
	Stroop commission	Placebo	0.47 ± 0.5	0.65 ± 0.9	0.59 ± 0.6	0.18 ± 0.9	0.12 ± 0.7	0.09	0.25	0.89	0.01	0.50
	Errors	Caffeine	0.53 ± 0.8	1.00 ± 1.0	0.82 ± 1.2	0.47 ± 1.1	0.29 ± 1.6					
		Matcha	0.94 ± 0.7	0.47 ± 0.6	0.77 ± 0.8	−0.47 ± 0.9 †	−0.18 ± 1.1					
	(C/B) × 100	Placebo	114 ± 14	112 ± 11	113 ± 8.8	−1.84 ± 15	−1.16 ± 15	0.68	0.51	0.26	0.66	0.44
		Caffeine	118 ± 15	114 ± 11	113 ± 10	−3.93 ± 13	−4.82 ± 18					
		Matcha	116 ± 10	116 ± 9.1	118 ± 12	−0.21 ± 14	1.46 ± 14					
	(A/B) × 100	Placebo	47.7 ± 5.9	48.9 ± 5.2	49.5 ± 5.7	1.23 ± 6.7	1.79 ± 7.3	0.24	0.66	0.20	0.19	0.69
		Caffeine	51.9 ± 7.9	51.7 ± 8.2	51.1 ± 7.3	−0.24 ± 7.8	−0.89 ± 9.0					
		Matcha	55.9 ± 16	50.4 ± 7.2	56.2 ± 10	−5.57 ± 12	0.28 ± 11					
SAT	Correct responses	Placebo	44.9 ± 7.5	47.0 ± 8.2	47.3 ± 7.1	2.12 ± 3.7	2.41 ± 3.8	0.45	0.09	0.11	0.06	0.89
		Caffeine	47.2 ± 6.4	52.6 ± 5.0	51.3 ± 4.3	5.41 ± 5.8	4.12 ± 6.6					
		Matcha	42.7 ± 13	50.0 ± 7.2	47.6 ± 7.7	7.29 ± 10	4.94 ± 11					
	Errors	Placebo	4.77 ± 3.3	3.88 ± 3.5	3.65 ± 3.1	−0.88 ± 2.3	−1.12 ± 2.3	0.22	0.13	0.18	0.82	0.84
		Caffeine	3.59 ± 3.5	1.88 ± 2.4	2.00 ± 1.5	−1.71 ± 3.2	−1.59 ± 3.9					
		Matcha	7.24 ± 7.8	3.41 ± 3.4	3.59 ± 3.1	−3.82 ± 7.7	−3.65 ± 7.1					
	Reaction time,	Placebo	1175 ±153	1152 ± 187	1147 ± 176	−23.4 ± 106	−27.9 ± 108	0.88	0.06	0.19	0.049	0.12
	correct (ms)	Caffeine	1148 ± 129	1033 ± 107 #	1053 ± 121	−115 ± 96 *	−94.8 ± 102					
		Matcha	1154 ± 197	1054 ± 145	1114 ± 148	−101 ± 152	−40.2 ± 190					
CPT	Correct responses	Placebo	39.8 ± 0.4	39.4 ± 1.6	39.4 ± 1.5	−0.47 ± 1.6	−0.41 ± 1.6	0.17	0.53	0.39	0.99	0.92
		Caffeine	39.9 ± 0.5	39.5 ± 1.7	39.8 ± 0.7	−0.35 ± 1.8	−0.06± 0.9					
		Matcha	40.0 ± 0.0	39.9 ± 0.2	39.9 ± 0.2	−0.06 ± 0.2	−0.06 ± 0.2					
	Omission errors	Placebo	0.18 ± 0.4	0.65 ± 1.6	0.59 ± 1.5	0.47 ± 1.6	0.41 ± 1.6	0.17	0.53	0.39	0.99	0.92
		Caffeine	0.12 ± 0.5	0.47 ± 1.7	0.18 ± 0.7	0.35 ± 1.8	0.06 ± 0.9					
		Matcha	0.00 ± 0.0	0.06 ± 0.2	0.06 ± 0.2	0.06 ± 0.2	0.06 ± 0.2					
	Commission errors	Placebo	0.24 ± 0.4	0.06 ± 0.2	0.24 ± 0.4	−0.18 ± 0.4	0.00 ± 0.6	0.95	0.57	0.61	0.73	0.77
		Caffeine	0.29 ± 0.7	0.12 ± 0.3	0.24 ± 0.4	−0.18 ± 0.7	−0.06 ± 0.7					
		Matcha	0.29 ± 0.6	0.18 ± 0.4	0.12 ± 0.3	−0.12 ± 0.6	−0.18 ± 0.6					
	Reaction time,	Placebo	480 ± 46	496 ± 53	482 ± 46	15.6 ± 41	2.00 ± 25	0.90	0.26	0.99	<0.01	0.47
	correct (ms)	Caffeine	484 ± 44	468 ± 38	480 ± 40	−16.1 ± 19 *	−4.35 ± 38					
		Matcha	515 ± 108	470 ± 38	480 ± 33	−45.5 ± 98 **	−35.8 ± 94					
FPCPT	Average response time,	Placebo	386 ± 64	385 ± 72	348 ± 42	−1.29 ± 53	−37.5 ± 46	0.36	0.11	0.56	0.51	0.99
(part 1)	correct (ms)	Caffeine	395 ± 82	364 ± 33	355 ± 41	−31.1 ± 60	−39.5 ± 61					
		Matcha	448 ± 137	403 ± 55	369 ± 55	−44.2 ± 117	−78.3 ± 137					
(part 2)	Correct responses	Placebo	6.00 ± 0.0	5.94 ± 0.2	5.59 ± 1.5	−0.06 ± 0.2	−0.41 ± 1.5	0.13	0.37	0.13	0.10	0.05
		Caffeine	5.82 ± 0.5	6.00 ± 0.0	6.00 ± 0.0	0.18 ± 0.5	0.18 ± 0.5					
		Matcha	6.00 ± 0.0	6.00 ± 0.0	6.00 ± 0.0	0.00 ± 0.0	0.00 ± 0.0					
	Average response time,	Placebo	447 ± 56	450 ± 65	428 ± 124	2.94 ± 42	−18.9 ± 119	0.48	0.51	0.85	0.03	0.31
	correct (ms)	Caffeine	464 ± 56	432 ± 40	441 ± 51	−31.8 ± 41 *	−22.7 ± 53					
		Matcha	473 ± 76	433 ± 43	444 ± 52	−40.4 ± 66	−29.7 ± 67					
	Incorrect responses	Placebo	0.24 ± 0.4	0.00 ± 0.0	0.24 ± 0.4	−0.24 ± 0.4	0.00 ± 0.7	0.06	0.06	0.43	0.048	0.12
		Caffeine	0.35 ± 0.6	0.29 ± 0.5	1.00 ± 4.1	−0.06 ± 0.6	0.65 ± 4.3					
		Matcha	0.00 ± 0.0	0.18 ± 0.4	1.06 ± 3.9	0.18 ± 0.4 *	1.06 ± 3.9					
	Average response time,	Placebo	118 ± 221	0.00 ± 0.0	92.9 ± 175	−118 ± 221	−25.4 ± 321	0.07	0.07	0.38	0.04	0.15
	incorrect (ms)	Caffeine	158 ± 265	122 ± 206	26.6 ± 110	−35.5 ± 229	−131 ± 302					
		Matcha	0.00 ± 0.0	72.1 ± 161	59.1 ± 135	72.1 ± 161 *	59.1 ± 135					
	Omission errors	Placebo	0.00 ± 0.0	0.06 ± 0.2	0.41 ± 1.5	0.06 ± 0.2	0.41 ± 1.5	0.13	0.37	0.13	0.10	0.05
		Caffeine	0.18 ± 0.5	0.00 ± 0.0	0.00 ± 0.0	−0.18 ± 0.5	−0.18 ± 0.5					
		Matcha	0.00 ± 0.0	0.00 ± 0.0	0.00 ± 0.0	0.00 ± 0.0	0.00 ± 0.0					

Values are presented as the mean ± standard deviation. Group (inter) analysis was performed using the Kruskal–Wallis test (KW) followed by a post-hoc Mann–Whitney U test at each time point to test for inter-group outcomes (control vs. caffeine, control vs. matcha, and caffeine vs. matcha). The *p*-value was determined using a Bonferroni correction of * *p* < 0.05/3 = 0.017; ** *p* < 0.01/3 = 0.0033 vs. placebo group; † *p* < 0.05/3 = 0.017, caffeine group vs. matcha group. (1) (2) (3) (4) (5) The numbers in brackets at −4 weeks, 0 weeks and 12 weeks corresponded to the numbers listed in the column with the *p*-value for the KW test. Time (intra) was analyzed using Dunnett’s test (T3) for intra-group comparisons, using the baseline condition as the reference (baseline vs. 0 W and baseline vs. 12 W). The *p*-value was determined using a Bonferroni correction of # *p* < 0.05/2 = 0.025. Abbreviations: ST, Stroop test; SAT, Shifting Attention test; CPT, Continuous Performance test; FPCPT, 4-part Continuous Performance test; KW, Kruskal–Wallis test; W, weeks.

**Table 6 nutrients-13-01700-t006:** Effect of matcha or caffeine intake on facial expression recognition-related tasks.

Task			−4 Weeks (1)	0 Weeks (2)	12 Weeks (3)	Change from Baseline		*p*-Values (KW Test)				
			(Baseline)	(Single Dose)		0 W (4)	12 W (5)	(1)	(2)	(3)	(4)	(5)
POET	Correct responses	Placebo	10.5 ± 1.0	10.4 ± 1.4	10.2 ± 1.7	−0.18 ± 1.1	−0.35 ± 1.7	0.09	0.98	0.74	0.11	0.14
		Caffeine	11.1 ± 0.8	10.4 ± 1.4	10.5 ± 1.3	−0.65 ± 1.1	−0.59 ± 1.2					
		Matcha	10.2 ± 1.3	10.5 ± 1.1	10.5 ± 1.7	0.29 ± 1.5	0.29 ± 1.6					
	Average reaction time	Placebo	1282 ± 161	1227 ± 183	1261 ± 205	−55.4 ± 185	−21.4 ± 149	0.13	0.07	0.14	0.76	0.77
	correct (ms)	Caffeine	1168 ± 140	1095 ± 151	1145 ± 166	−72.2 ± 108	−23.0 ± 176					
		Matcha	1228 ± 185	1148 ± 151	1220 ± 127	−80.2 ± 186	−7.65 ± 189					
	Omission errors	Placebo	1.47 ± 1.0	1.65 ± 1.4	1.82 ± 1.7	0.18 ± 1.1	0.35 ± 1.7	0.09	0.98	0.74	0.11	0.14
		Caffeine	0.94 ± 0.8	1.59 ± 1.4	1.53 ± 1.3	0.65 ± 1.1	0.59 ± 1.2					
		Matcha	1.77 ± 1.3	1.47 ± 1.1	1.47 ± 1.7	−0.29 ± 1.5	−0.29 ± 1.6					
	Commission errors	Placebo	3.35 ± 1.9	2.53 ± 1.8	2.12 ± 1.5	−0.82 ± 1.2	−1.24 ± 1.3	0.39	0.94	0.69	0.58	0.88
		Caffeine	2.59 ± 2.3	2.29 ± 1.8	1.82 ± 2.0	−0.29 ± 2.0	−0.77 ± 2.9					
		Matcha	3.41 ± 3.1	2.53 ± 2.5	2.12 ± 1.9	−0.88 ± 2.3	−1.29 ± 2.5					
Positive	Correct hits	Placebo	5.77 ± 0.4	5.35 ± 0.9	5.29 ± 1.0	−0.41 ± 1.1	−0.47 ± 0.9	0.76	0.57	0.37	0.30	0.23
Emotions		Caffeine	5.65 ± 0.5	5.12 ± 1.2	5.41 ± 1.0	−0.53 ± 0.9	−0.24 ± 0.8					
		Matcha	5.65 ± 0.6	5.59 ± 0.6	5.65 ± 0.9	−0.06 ± 0.8	0.00± 0.7					
	Reaction time (ms)	Placebo	1285 ± 184	1227 ± 184	1234 ± 223	−58.2 ± 227	−51.2 ± 186	0.08	0.09	0.35	0.61	0.53
		Caffeine	1136 ± 185	1104 ± 163	1141 ± 207	−32.6 ± 155	4.53 ± 189					
		Matcha	1231 ± 193	1139 ± 144	1168 ± 127	−92.4 ± 197	−63.2 ± 182					
Negative	Correct hits	Placebo	4.77 ± 0.8	5.00 ± 1.0	4.88 ± 1.2	0.24 ± 0.9	0.12 ± 1.6	0.04	0.55	0.91	0.46	0.27
Emotions		Caffeine	5.41 ± 0.8	5.29 ± 0.8	5.06 ± 1.0	−0.12 ± 0.8	−0.35 ± 1.2					
		Matcha	4.59 ± 1.2	4.94 ± 1.0	4.88 ± 1.2	0.35 ± 1.3	0.29 ± 1.4					
	Reaction time (ms)	Placebo	1278 ± 203	1229 ± 223	1289 ± 212	−48.8 ± 193	11.5 ± 193	0.56	0.18	0.04	0.87	0.32
		Caffeine	1204 ± 169	1096 ± 182	1142 ± 183	−108 ± 212	−62.2 ± 253					
		Matcha	1227 ± 227	1164 ± 206	1291 ± 174	−63.1 ± 249	63.8 ± 274					

Values are presented as the mean ± standard deviation. Group (inter) analysis was performed using the Kruskal–Wallis test (KW). (1) (2) (3) (4) (5) The numbers in brackets at −4 weeks, 0 weeks and 12 weeks corresponded to the numbers listed in the column with the *p*-value for the KW test. Time (intra) was analyzed using Dunnett’s test (T3) for intra-group comparisons, using the baseline condition as the reference (baseline vs. 0 W and baseline vs. 12 W). Abbreviations: KW, Kruskal–Wallis test; POET, Perception of Emotions test; W, weeks.

**Table 7 nutrients-13-01700-t007:** Effect of matcha or caffeine intake on working memory-related tasks.

Task			−4 Weeks (1)	0 Weeks (2)	12 Weeks (3)	Change from Baseline		*p*-Values (KW Test)				
			(Baseline)	(Single Dose)		0 W (4)	12 W (5)	(1)	(2)	(3)	(4)	(5)
FPCPT	Correct responses	Placebo	15.1 ± 1.8	15.5 ± 0.9	15.3 ± 1.0	0.35 ± 2.1	0.18 ± 1.8	0.99	0.88	0.41	0.90	0.87
(part 3)		Caffeine	15.1 ± 1.7	15.5 ± 1.0	15.6 ± 0.7	0.41 ± 1.6	0.53 ± 1.9					
		Matcha	15.4 ± 0.9	15.6 ± 0.6	15.5 ± 0.8	0.24 ± 0.8	0.12 ± 1.3					
	Average response time,	Placebo	590 ± 102	572 ± 126	614 ± 131	−18.2 ± 7.9	24.2 ± 72	0.31	0.51	0.06	0.80	0.41
	correct (ms)	Caffeine	543 ± 101	528 ± 93	529 ± 107	−15.2 ± 68	−13.5 ± 126					
		Matcha	565 ± 115	557 ± 111	544 ± 77	−8.24 ± 57	−20.9 ± 108					
	Incorrect responses	Placebo	0.00 ± 0.0	0.06 ± 0.2	0.06 ± 0.2	0.06 ± 0.2	0.06 ± 0.2	0.37	0.59	0.42	0.58	0.81
		Caffeine	0.00 ± 0.0	0.18 ± 0.4	0.06 ± 0.2	0.18 ± 0.4	0.06 ± 0.2					
		Matcha	0.06 ± 0.2	0.18 ± 0.5	0.18 ± 0.4	0.12 ± 0.6	0.12 ± 0.5					
	Average response time,	Placebo	0.00 ± 0.0	42.1 ± 173	57.1 ± 235	42.1 ± 173	57.1 ± 235	0.37	0.56	0.44	0.52	0.83
	incorrect (ms)	Caffeine	0.00 ± 0.0	139 ± 348	61.5 ± 254	139 ± 348	61.5 ± 254					
		Matcha	71.2 ± 293	65.9 ± 189	216 ± 633	−5.24 ± 363	144 ± 720					
	Omission errors	Placebo	0.88 ± 1.8	0.53 ± 0.9	0.71 ± 1.0	−0.35 ± 2.1	−0.18 ± 1.8	0.99	0.88	0.41	0.90	0.87
		Caffeine	0.88 ± 1.7	0.47 ± 1.0	0.35 ± 0.7	−0.41 ± 1.6	−0.53 ± 1.9					
		Matcha	0.59 ± 0.9	0.35 ± 0.6	0.47 ± 0.8	−0.24 ± 0.8	−0.12 ± 1.3					
(part 4)	Correct responses	Placebo	10.7 ± 2.4	11.5 ± 3.0	12.3 ± 2.8	0.82 ± 3.0	1.59 ± 2.2	0.03	0.40	0.56	0.27	0.05
		Caffeine	12.6 ± 2.3	12.8 ± 2.1	13.1 ± 2.0	0.18 ± 2.6	0.53 ± 2.3					
		Matcha	12.7 ± 2.6	12.2 ± 2.0	12.4 ± 2.2	−0.47 ± 2.6	−0.35 ± 2.5					
	Average response time,	Placebo	745 ± 124	748 ± 143	712 ± 173	2.94 ± 132	−33.2 ± 149	0.21	0.04	0.16	0.50	0.60
	correct (ms)	Caffeine	667 ± 146	618 ± 145 *	625 ± 123	−48.6 ± 120	−41.4 ± 117					
		Matcha	722 ± 119	692 ± 140	646 ± 104	−29.8 ± 123	−76.5 ± 124					
	Incorrect responses	Placebo	1.65 ± 2.0	1.12 ± 0.9	1.12 ± 1.9	−0.53 ± 1.6	−0.53 ± 2.3	0.54	0.51	0.19	0.34	0.30
		Caffeine	1.41 ± 1.2	1.82 ± 1.7	1.47 ± 1.1	0.41 ± 1.7	0.06± 1.0					
		Matcha	1.00 ± 1.2	1.47 ± 1.5	1.65 ± 1.5	0.47± 1.5	0.65± 1.5					
	Average response time,	Placebo	439 ± 443	795 ± 546	511 ± 553	356 ± 642	71.8 ± 747	0.15	0.22	0.57	0.02	0.38
	incorrect (ms)	Caffeine	760 ± 568	515 ± 392	623 ± 484	−245 ± 555 *	−137 ± 725					
		Matcha	534 ± 583	615 ± 483	687 ± 457	81.1 ± 568	153 ± 779					
	Omission errors	Placebo	5.29 ± 2.4	4.47 ± 3.0	3.71 ± 2.8	−0.82 ± 3.0	−1.59 ± 2.2	0.03	0.40	0.56	0.27	0.05
		Caffeine	3.41 ± 2.3	3.24 ± 2.1	2.88 ± 2.0	−0.18± 2.6	−0.53 ± 2.3					
		Matcha	3.29 ± 2.6	3.77 ± 2.0	3.65 ± 2.2	0.47± 2.6	0.35± 2.5					

Values are presented as the mean ± standard deviation. Group (inter) analysis was performed using the Kruskal–Wallis test (KW) followed by a post-hoc Mann–Whitney U test at each time point to test for inter-group outcomes (control vs. caffeine, control vs. matcha, and caffeine vs. matcha). The *p*-value was determined using a Bonferroni correction of * *p* < 0.05/3 = 0.017. (1) (2) (3) (4) (5) The numbers in brackets at −4 weeks, 0 weeks and 12 weeks corresponded to the numbers listed in the column with the *p*-value for the KW test. Time (intra) was analyzed using Dunnett’s test (T3) for intra-group comparisons, using the baseline condition as the reference (baseline vs. 0 W and baseline vs. 12 W). Abbreviations: FPCPT, 4-part Continuous Performance test; KW, Kruskal–Wallis test; W, weeks.

**Table 8 nutrients-13-01700-t008:** Effect of matcha or caffeine intake on visual information processing-related tasks.

Task			−4 Weeks (1)	0 Weeks (2)	12 Weeks (3)	Change from Baseline		*p*-Values (KW Test)				
			(Baseline)	(Single Dose)		0 W (4)	12 W (5)	(1)	(2)	(3)	(4)	(5)
SDC	Correct responses	Placebo	57.5 ± 10	58.1 ± 8.6	59.6 ± 8.9	0.53 ± 5.3	2.06 ± 5.6	0.46	0.23	0.22	0.11	0.46
		Caffeine	59.1 ± 6.9	62.2 ± 5.4	63.4 ± 5.0	3.12 ± 3.7	4.24 ± 3.7					
		Matcha	55.4 ± 9.0	59.2 ± 7.2	59.4 ± 7.8	3.88 ± 6.1	4.00 ± 8.4					
	Errors	Placebo	0.53 ± 0.9	1.18 ± 2.3	0.53 ± 1.0	0.65 ± 2.4	0.00 ± 1.5	0.22	0.82	0.55	0.16	0.78
		Caffeine	0.53 ± 0.6	0.41 ± 0.6	0.41 ± 0.6	−0.12 ± 0.5	−0.12 ± 0.9					
		Matcha	1.71 ± 4.0	0.88 ± 1.8	0.59 ± 0.6	−0.82 ± 3.7	−1.12 ± 4.0					
NVRT	Correct responses	Placebo	9.77 ± 1.8	10.6 ± 2.0	9.94 ± 1.8	0.82 ± 1.7	0.18 ± 1.9	0.37	0.22	0.91	0.27	0.21
		Caffeine	10.3 ± 2.1	10.2 ± 2.0	9.88 ± 2.0	−0.12 ± 1.8	−0.41 ± 2.4					
		Matcha	9.35 ± 2.0	9.18 ± 3.0	10.2 ± 2.4	−0.18 ± 2.7	0.82 ± 2.7					
	Average reaction time,	Placebo	4255 ± 1132	4406 ± 825	4338 ± 743	151 ± 968	82.9 ± 1022	0.66	0.38	0.94	0.06	0.53
	correct (ms)	Caffeine	4566 ± 926	3961 ± 1153	4238 ± 782	−605 ± 1202	−327 ± 1021					
		Matcha	4506 ± 1075	4030 ± 1001	4297 ± 1043	−476 ± 1006	−209 ± 1252					
	Commission errors	Placebo	4.82 ± 2.0	4.00 ± 1.9	4.65 ± 2.1	−0.82 ± 1.5	−0.18 ± 2.0	0.53	0.16	0.98	0.08	0.43
		Caffeine	4.18 ± 2.2	4.65 ± 2.1	4.77 ± 2.3	0.47± 1.9	0.59 ± 2.6					
		Matcha	4.94 ± 2.2	5.65 ± 3.2	4.59 ± 2.6	0.71± 2.8	−0.35 ± 3.1					
	Omission errors	Placebo	0.41 ± 0.5	0.41 ± 0.6	0.41 ± 0.6	0.00± 0.9	0.00 ± 0.6	0.75	0.34	0.67	0.40	0.41
		Caffeine	0.53 ± 0.6	0.18 ± 0.4	0.35 ± 0.5	−0.35 ± 0.7	−0.18 ± 0.6					
		Matcha	0.71 ± 0.9	0.18 ± 0.4	0.24 ± 0.4	−0.53 ± 1.0	−0.47 ± 1.0					

Values are presented as the mean ± standard deviation. Group (inter) analysis was performed using the Kruskal–Wallis test (KW). (1) (2) (3) (4) (5) The numbers in brackets at −4 weeks, 0 weeks and 12 weeks corresponded to the numbers listed in the column with the *p*-value for the KW test. Time (intra) was analyzed using Dunnett’s test (T3) for intra-group comparisons, using the baseline condition as the reference (baseline vs. 0 W and baseline vs. 12 W). Abbreviations: SDC, Symbol Digit Coding test; NVRT, Non-Verbal Reasoning test; KW, Kruskal–Wallis test; W, weeks.

**Table 9 nutrients-13-01700-t009:** Effect of matcha or caffeine intake on motor function-related tasks.

Task			−4 Weeks (1)	0 Weeks (2)	12 Weeks (3)	Change from Baseline		*p*-Values (KW Test)				
			(Baseline)	(Single Dose)		0 W (4)	12 W (5)	(1)	(2)	(3)	(4)	(5)
FTT	Right taps,	Placebo	60.9 ± 5.2	61.1 ± 4.6	59.8 ± 5.4	0.12 ± 1.7	−1.12 ± 4.5	0.50	0.93	0.29	0.45	0.39
	average	Caffeine	58.8 ± 5.2	60.6 ± 4.0	56.6 ± 7.9	1.82 ± 3.6	−2.12 ± 6.4					
		Matcha	59.4 ± 6.2	60.5 ± 5.6	59.5 ± 5.6	1.12 ± 4.0	0.12 ± 4.9					
	Left taps,	Placebo	57.1 ± 5.3	57.1 ± 5.8	54.9 ± 6.1	0.00 ± 3.0	−2.18 ± 3.5	0.12	0.18	0.14	0.37	0.51
	average	Caffeine	53.4 ± 5.5	53.4 ± 6.5	50.6 ± 7.0	0.00 ± 3.8	−2.76 ± 5.3					
		Matcha	54.1 ± 5.9	55.6 ± 5.3	53.6 ± 5.9	1.59 ± 2.9	−0.47 ± 3.6					

Values are presented as the mean ± standard deviation. Group (inter) analysis was performed using the Kruskal–Wallis test (KW). (1) (2) (3) (4) (5) The numbers in brackets at −4 weeks, 0 weeks and 12 weeks corresponded to the numbers listed in the column with the *p*-value for the KW test. Time (intra) was analyzed using Dunnett’s test (T3) for intra-group comparisons, using the baseline condition as the reference (baseline vs. 0 W and baseline vs. 12 W). Abbreviations: FTT, Finger Tapping test; KW, Kruskal–Wallis test; W, weeks.

**Table 10 nutrients-13-01700-t010:** Effect of matcha or caffeine intake on dementia-related blood biomarkers.

			−4 Weeks (1)	12 Weeks (2)	Change from Baseline (3)	*p*-Values (KW Test)		
		n	(Baseline)			(1)	(2)	(3)
Plasma Aβ (1–40; pg/mL)	Placebo	11	251 ± 56	353 ± 369	102 ± 385	0.26	0.96	0.56
	Caffeine	13	275 ± 109	263 ± 66	−11.9 ± 77			
	Matcha	9	415 ± 392	255 ± 59	−161 ± 410			
Plasma Aβ (1–42; pg/mL)	Placebo	11	16.2 ± 18	35.3 ± 58	19.1 ± 54	0.64	0.046	0.35
	Caffeine	13	13.3 ± 10	35.6 ± 69	22.3 ± 71			
	Matcha	9	17.5 ± 20	10.6 ± 4.4	−6.94 ± 22			
Aβ (1–42)/Aβ (1–40)	Placebo	11	0.066 ± 0.06	0.085 ± 0.09	0.019 ± 0.05	0.69	0.12	0.36
	Caffeine	13	0.054 ± 0.04	0.128 ± 0.22	0.074 ± 0.23			
	Matcha	9	0.040 ± 0.01	0.042 ± 0.02	0.002 ± 0.02			
Plasma sAPPα (ng/mL)	Placebo	16	6.51 ± 1.5	9.21 ± 3.3 ##	2.70 ± 3.1	0.82	0.16	0.16
	Caffeine	17	6.16 ± 1.7	9.83 ± 5.8 ##	3.67 ± 5.4			
	Matcha	17	6.28 ± 2.1	7.08 ± 2.1 ##	0.80 ± 1.8			
Plasma APP770 (ng/mL)	Placebo	17	27.7 ± 11	29.1 ± 15	1.49 ± 12	0.52	0.21	0.13
	Caffeine	17	33.1 ± 15	33.5 ± 27	0.38 ± 19			
	Matcha	17	28.8 ± 11	20.9 ± 5.7 ##	−7.93 ± 10			
Serum BDNF (ng/mL)	Placebo	17	39.0 ± 13	30.6 ± 9.1 ##	−8.40 ± 6.1	0.89	0.99	0.78
	Caffeine	17	41.0 ± 13	31.1 ± 11 ##	−10.0 ± 5.7			
	Matcha	17	38.3 ± 11	29.5 ± 7.4 ##	−8.87 ± 5.6			

Values are presented as the mean ± standard deviation. Group (inter) analysis was performed using the Kruskal–Wallis test (KW). (1) (2) (3) The numbers in brackets at −4 weeks and 12 weeks corresponded to the numbers listed in the *p*-value for the KW test. Time (intra) was analyzed using Dunnett’s test (T3) for intra-group comparisons, using the baseline condition as the reference (baseline vs. 0 W and baseline vs. 12 W); ## *p* < 0.01. Abbreviations: Aβ, amyloid β; sAPPα, secreted form of amyloid precursor protein α; APP, amyloid precursor protein; BDNF, brain-derived neurotrophic factor; KW, Kruskal–Wallis test.

## Data Availability

The data presented in this study are available on request from the corresponding author. The data are not publicly available due to legal restrictions.

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
