# Peer review of "Effects of Daily Matcha and Caffeine Intake on Mild Acute Psychological Stress-Related Cognitive Function in Middle-Aged and Older Adults: A Randomized Placebo-Controlled Study"

_nutrients, 2021, doi:10.3390/nu13051700_

Round 1

Reviewer 1 Report

The paper ‘Effects of daily matcha and caffeine intake on mild acute psychological stress-related cognitive function in middle-aged and older adults: A randomized placebo-controlled study’ is clearly presented. Authors have shown that caffeine improves attentional function during stress loading, and a single dose of matcha reduced the reaction time in cognitive function tests.

Major comments:

1) Please, provide better patient group description in “methods section”. Nutrients concentrations between all groups must be in a table format.

2) Please, improve figure and table legends for all figures and tables.

Minor comments:

1) The first sentence in the abstract is too simplistic. Please rewrite it.

Reviewer 2 Report

The manuscript by Yoshitake Baba et al. entitled " Effects of daily matcha and caffeine intake on mild acute psychological stress-related cognitive function in middle-aged and older adults: A randomized placebo-controlled study". This paper presents randomized placebo-controlled studies aimed at comparing the effects of caffeine and matcha and explaining the differences between these effects. The issue undertaken by the Authors is interesting and the results are novel and noteworthy. However, this manuscript suffers from some minor problems that are listed below.

  1. The Authors are inconsequent in the use of abbreviations. At first use the abbreviation should be explained and then used consistently throughout the manuscript.
  2. Introduction: The Authors mention that caffeine acts via blocking A2A adenosine receptors, but this is not the only mechanism of caffeine's action. It would be worthwhile to describe the mechanism of action of caffeine in more detail.
  3. Aim of studies: The purpose of the work presented in the Introduction section requires clarification. It does not follow from it clear that the effects of caffeine and matcha were compared.
  4. Subsection 3.4. Blood Biomarkers and Table 9: It is unclear whether the biomarker levels were marked in plasma or serum. The Authors interchangeably use "plasma" and "serum".
  5. Table 9: Plasma Aβ (1-42; pg/mL) p-values [2] = 0.046, but the Authors indicated that there was no statistical significance for any of the marked biomarkers.
  6. Discussion section: l. 360 “Authors” should be removed.

Round 2

Reviewer 1 Report

I recommend to accept this paper in present form

Author Response

Thank you very much for peer review.

We are thankful for the time and energy you expended.

I have fixed the following points with the advice of the Academic Editor:

I moved the tables to the result section (Table4–9).

I modified the table legends (Table4–10 and Fig 2).

In addition, this manuscript has been proofread in English.

Matcha is an interesting food that has the effect of improving cognitive function. I want the world to know the positive effects of matcha.

Best regards,

Yoshitake Baba